# An Engineered Laccase from *Fomitiporia mediterranea* Accelerates Lignocellulose Degradation

**DOI:** 10.3390/biom14030324

**Published:** 2024-03-08

**Authors:** Le Thanh Mai Pham, Kai Deng, Hemant Choudhary, Trent R. Northen, Steven W. Singer, Paul D. Adams, Blake A. Simmons, Kenneth L. Sale

**Affiliations:** 1Department of Bioresource & Environmental Security, Sandia National Laboratories, 7011 East Avenue, Livermore, CA 94551, USA; 2Deconstruction Division, Joint BioEnergy Institute, 5885 Hollis Street, Emeryville, CA 94608, USA; 3Department of Biomaterials & Biomanufacturing, Sandia National Laboratories, 7011 East Avenue, Livermore, CA 94551, USA; 4Technology Division, Joint BioEnergy Institute, 5885 Hollis Street, Emeryville, CA 94608, USA; 5Environmental Genomics and Systems Biology Divison, Lawrence Berkeley National Laboratory, 1 Cyclotron Road, Berkeley, CA 94720, USA; 6Biological Systems and Engineering Division, Lawrence Berkeley National Laboratory, 1 Cyclotron Road, Berkeley, CA 94720, USA; 7Molecular Biophysics and Integrated Bioimaging Division, Lawrence Berkeley National Laboratory, 1 Cyclotron Road, Berkeley, CA 94720, USA; 8Department of Bioengineering, University of California, Berkeley, CA 94704, USA; 9Department of Biosecurity and Bioassurance, Sandia National Laboratories, 7011 East Avenue, Livermore, CA 94551, USA

**Keywords:** lignin, laccase, nanostructure-initiator mass spectrometry (NIMS), *Komagataella pastoris* expression, *Fomitiporia mediterranea*

## Abstract

Laccases from white-rot fungi catalyze lignin depolymerization, a critical first step to upgrading lignin to valuable biodiesel fuels and chemicals. In this study, a wildtype laccase from the basidiomycete *Fomitiporia mediterranea* (Fom_lac) and a variant engineered to have a carbohydrate-binding module (Fom_CBM) were studied for their ability to catalyze cleavage of β-O-4′ ether and C–C bonds in phenolic and non-phenolic lignin dimers using a nanostructure-initiator mass spectrometry-based assay. Fom_lac and Fom_CBM catalyze β-O-4′ ether and C–C bond breaking, with higher activity under acidic conditions (pH < 6). The potential of Fom_lac and Fom_CBM to enhance saccharification yields from untreated and ionic liquid pretreated pine was also investigated. Adding Fom_CBM to mixtures of cellulases and hemicellulases improved sugar yields by 140% on untreated pine and 32% on cholinium lysinate pretreated pine when compared to the inclusion of Fom_lac to the same mixtures. Adding either Fom_lac or Fom_CBM to mixtures of cellulases and hemicellulases effectively accelerates enzymatic hydrolysis, demonstrating its potential applications for lignocellulose valorization. We postulate that additional increases in sugar yields for the Fom_CBM enzyme mixtures were due to Fom_CBM being brought more proximal to lignin through binding to either cellulose or lignin itself.

## 1. Introduction

Lignin is a complex organic polymer that plays a crucial role in the structure of plant cell walls. It is one of the three main components of plant cell walls: cellulose and hemicellulose. While cellulose provides strength and rigidity to the cell wall, and hemicellulose contributes to its flexibility, lignin acts as a binding agent, providing additional support and resistance to decay [1]. Lignin has great potential in the biofuel industry, but challenges remain, such as developing cost-effective and scalable processes for lignin depolymerization and conversion to valuable products. Ongoing research and advancements in biotechnology and chemical engineering are critical for unlocking lignin’s full potential in producing sustainable biofuels [2].

Lignin degradation in nature is primarily carried out by various microorganisms, including fungi and bacteria, which produce a variety of enzymes to break down the complex structure of lignin. Three key enzymes, Laccases, Lignin Peroxidases (LiP), and Manganese Peroxidases (MnP), involved in lignin degradation are ligninolytic enzymes, and the process is often referred to as ligninolysis. These enzymes work together in a coordinated manner to depolymerize lignin into smaller fragments that microorganisms can further metabolize for energy and carbon. It is important to note that the specific enzymes and mechanisms involved can vary among different microorganisms, and some species may produce a combination of these enzymes to degrade lignin efficiently. Studying these natural lignin-degrading enzyme systems is critical to gaining insights into how they can be harnessed for industrial applications, such as biofuel production and bioremediation. The complex interactions among the agents in secretomes can lead to difficulties in elucidating the mechanisms of lignin-degrading enzymes and make it particularly difficult to compare enzymes from either the same enzyme family or the same fungus. Therefore, instead of mixed secretomes of ligninolytic enzymes, heterologous expression of individual genes, purification of the resulting enzymes, and quantification of bond-breaking events is a valuable approach to studying and comparing the structure–function relationships of these essential enzymes and to building potent enzyme mixtures for efficient lignin depolymerization. 

Laccases (EC 1.10.3.2) are copper-containing enzymes capable of oxidizing electron-rich organic and inorganic substrates using molecular oxygen as an electron acceptor and are found in plants, fungi, and bacteria [3]. In fungi, they play critical roles in several physical functions, such as morphogenesis, fungal plant pathogen/host interaction, stress defense, and lignin degradation. Due to their ability to oxidize many substrates, laccases have applications in various industries, including pulp and paper processing, textile dyeing, and wastewater treatment. Additionally, laccases have been used in biotechnological processes, such as the modification of lignin for biofuel production and the degradation of environmental pollutants. The versatility of laccases makes them valuable tools in both natural ecosystems and industrial applications. *Fomitiporia mediterranea* is a polypore fungal species that grows on olive trees [4] and has been associated with esca in grapevines and their roots [5,6]. Functionally, *F. mediterranea* is a white rot-causing basidiomycete that secretes both cellulolytic enzymes that catalyze depolymerization of cellulose and hemicellulose and ligninolytic enzymes, including laccases and peroxidases, which catalyze depolymerization of lignin. It is reported that incubation of purified laccase from the secretomes of *F. mediterranea* with natural phenolic and polyphenolic compounds resulted in the oxidation of both compounds [6]. To date, studies aimed at the recombinant expression of laccases from *F. mediterranea,* and subsequent characterization of the ability of the purified laccase to catalyze the breaking of bonds commonly found in lignin and their optimal reaction conditions, have not been reported, suggesting further research is needed to explore the potential of *F. mediterranea* laccase in processes requiring depolymerization of lignin and polysaccharides. 

To this end, the primary aim of this work was to characterize the catalytic performance of a heterologously expressed (*Komagataella pastoris*) and purified laccase from the white rot-causing basidiomycete *F. mediterranea* (Fom_lac) by quantifying β-O-4′ ether, C_α_-C_β_, and C_α_-C_1_ bond cleavage products and by-products produced from C_α_-oxidation and polymerization of reaction products using a nanostructure-initiator mass spectrometry (NIMS) assay with phenolic and non-phenolic lignin-like model compounds [7]. Various factors influence lignin degradation, and pH is one of the critical parameters that can significantly affect this process. Indeed, numerous studies have explored the effect of pH on laccase-catalyzed lignin degradation [8,9]. Both the effects of pH and the presence of the reaction mediator hydroxybenzotriole (HBT) were investigated. A secondary aim was to evaluate the utility of Fom_lac and an engineered variant of Fom_lac to which a carbohydrate-binding module (CBM) was fused to its C-terminus (Fom_CBM) to enhance saccharification yields from dry milled and cholinium ([Ch]+) lysinate ([Lys]−) ([Ch][Lys]) pretreated pine (*Pinus radiata*) by measuring glucose and xylose yields. Pine was chosen for these experiments because it is a highly recalcitrant feedstock with a high lignin content, making it an ideal substrate for studies aimed at exploring the utility, in terms of increased monosaccharide yields, of adding lignin-modifying enzymes such as laccases to the cellulolytic enzyme mixtures. Pretreatment with ionic liquid, particularly [Ch][Lys], was chosen because [Ch][Lys] pretreatment produces a less recalcitrant biomass [10]. This study will investigate whether the addition of laccases to hydrolytic enzyme mixtures can further improve sugar yields over what was achieved by a very efficient pretreatment solvent. 

## 2. Materials and Methods

### 2.1. Cloning and Expression of Laccase Variants

The laccase gene from the genome of *F. mediterranea* Fom_Lac was identified in the JGI—MycoCosm database (Protein ID: 127515, https://mycocosm.jgi.doe.gov/Fomme1/Fomme1.home.html, accessed on 31 May 2020) as a recently added entry [11]. The CBM from the *Trichoderma reesei* exoglucanase 1 (Uniprok protein ID: P62694) was selected from a pool of eleven candidate CBMs, as described in the Simulation Methods Section, and fused to the C-terminus of Fom_lac. Expression and purification of both Fom_lac and Fom_CBM followed methods reported in a previous publication [8]. Briefly, codon-optimized genes (Genscript Co., Piscataway, NJ, USA) with recognition sites for XhoI and EcoRI restriction sites were cloned into the pPICZαA vector from Invitrogene™ (Life Technologies, Carlsbad, CA, USA), linearized with PmeI and SacI, and transformed into *K. pastoris* X33 strain, as described in the *Pichia* Expression Kit User Manual (Life Technologies, USA). Transformants were grown at 30 °C on yeast extract peptone dextrose medium with sorbitol (YPDS) agar plates containing 100 µg/mL Zeocin™ antibiotic (Thermo Fisher Scientific, Waltham, MA, USA). Clones from positive colonies grown on YPDS agar plates containing 1000 µg/mL Zeocin™ were selected and grown in buffered complex methanol medium (BMMY) overnight at 30 °C at 200 rpm. The *K. pastoris* was grown to an OD_600_ of 0.6 in 250 mL flasks with 50 mL BMMY. Cultures were shaken at 30 °C and 200 rpm for 3–5 days, fed daily with a 1% (*v*/*v*) methanol solution and 1 mM CuSO_4,_ and stopped when the laccase activity, measured as oxidation of ABTS substrate, reached saturation levels. 

### 2.2. Purification of Recombinant Fom_lac Enzyme

Enzyme purification followed procedures described previously [8]. Briefly, culture supernatants were centrifuged (9000 rpm for 15 min), clarified with 0.2 µm membrane filtration, concentrated 15X using Amicon^®^ Ultra-50-kDa Centrifugal Filter Units (MilliporeSigma, Burlington, MA, USA), dialyzed overnight through a 10-kDa membrane against 100 mM sodium acetate (pH 3.0) at 4 °C, and again dialyzed overnight against 10 mM sodium acetate (pH 6.0) at 4 °C. The precipitate was removed by centrifugation at 13,000 rpm for 15 min and filtered through 0.2 µm membrane filtration, which was then loaded onto the Hitrap-Q XL (5 mL) column with an AKTA FPLC (G.E. Healthcare, Chalfont Saint Giles, UK) purification column. Active fractions were pooled using stepwise gradients of buffer A (10 mM sodium acetate, pH 6.0) and buffer B (500 mM sodium acetate, pH 6.0), and fractions with the highest activity towards ABTS oxidation were collected. 

### 2.3. Enzyme Reactions with the Fluorous-Tagged Phenolic/Nonphenolic β-O-4 Linked Model Compound

The phenolic β-O-4 aryl ether lignin-like model compounds, NIMS-tagged guaiacylglycerol-beta-guaiacyl ether (GGE) and NIMS-tagged veratrylglycerol-beta-guaiacyl ether (VGE), were synthesized according to a previously established protocol [7]. Enzyme reactions with the NIMS-tagged GGE (1 mM) were performed at pH 2.0–10.0 and in the absence and presence of 20 mM of 1-hydrobenzotriazole (HBT) as a mediator. The reaction was stopped after 3 h, and analysis of reaction products was performed using nanostructure-initiator mass spectrometry (NIMS) as previously described [7]. As negative control, substrate solution in the absence of laccase was treated in the same way. 

### 2.4. Biomass Feedstock Preparation

Dry-milled pine (*Pinus radiata*) and 80% [Ch][Lys]-pretreated pine were used as feedstocks for enzymatic saccharification reactions used to study synergy among laccases, cellulases, and hemicellulases. Dry-milled pine was separated into fractions using a sieve with an aperture size of 2 mm. The [Ch][Lys] pretreated pine was generated as reported previously with slight modifications [12]. Briefly, pine was pretreated in a 2:8 ratio by weight of pine to the ionic liquid [Ch][Lys] in a 1 L Parr 4520 series benchtop reactor (Parr Instrument Company, model 4871, Moline, IL, USA) for 3 h at 140 °C with stirring at 80 rpm using a three-arm self-centering anchor with Polytetrafluoroethylene (PTFE) wiper blades. The process was controlled and monitored using the Parr ‘Instruments’ model 4871 process controller and a model 4875 power controller. After 3 h, the pretreated slurry was cooled down to room temperature by removing the heating jacket. Pretreated pine was washed with DI water until the pH of the washing water was neutral and it was finally freeze-dried to obtain a free-flowing solid residue.

### 2.5. Enzymatic Saccharification Reactions

A 9:1 (*v*/*v*) mixture of cellulase (Cellic^®^ Ctec3, 1853 BHU-2-HS g^−1^, 1.212 g mL^−1^) (Novozymes North America, Franklinton, NC, USA) and hemicellulase (Htec3 NS 22244, 1760 FXU g^−1^, 1.210 g mL^−1^) (Novozymes) was used for all saccharification reactions. Reactions were carried out using a 2.5% biomass loading and an enzyme dose of 10 mg protein per 1 g biomass, and they were supplemented with 0.02% sodium azide to prevent microbial contamination [10]. The synergistic effect of laccases with the CTec3/HTec3 enzyme mixture was studied by adding 5 uM Fom_lac or Fom_CBM and 5 mM HBT for reactions with the mediator. Reactions were run for 72 h at pH 5.5 and 60 °C, and hydrolysates were separated from the residual solids by centrifugation followed by filtration through 0.2 μm sterile filter units. 

Glucose and xylose yields were quantified by HPLC using an Agilent HPLC 1260 infinity system (Agilent Technologies, Santa Clara, CA, USA) with an Aminex™ HPX-87 H column (Bio-Rad, Hercules, CA, USA) and a Refractive Index detector. The column was eluted using a 4 mM sulfuric acid solution, a 0.6 mL/min rate, and a column temperature of 60 °C. Standards for quantification were obtained from Sigma-Aldrich (St. Louis, MO, USA). As negative control, biomass in the absence of enzymes was treated in the same way.

### 2.6. Simulation Methods 

Gibbs free energy of oxidation of the lignin dimers was calculated and AIMD simulations of the cationic radical intermediate and bond dissociation energies for all bond types in lignin dimers were performed as previously described [13]. Calculations were performed in the Gaussian 09 software package [14] using unrestricted density functional theory (DFT) with the Becke three-parameter Lee–Yang–Parr hybrid exchange–correlation functional (B3LYP) [15,16] and 6-311G** basis set [17,18] and the implicit solvation model based on density (SMD) [19]. 

The TeraChem quantum chemistry package (Petachem LLC, Los Altos, CA, USA) was used to perform Ab initio Molecular dynamic (AIMD) simulations [20,21,22,23]. The ab initio calculations in these simulations were performed using unrestricted density functional theory (DFT) with the long-range corrected ωPBEh exchange-correlation functional [24] and 6–31 g basis set. A bond was determined to be broken when the separation distance between two atoms exceeded the bond length of the corresponding bond in the initial structure, e.g., breaking of the C_a_-OH bond, C_α_-C_β_Bond, β-O-4′ ether bond, and C_α_-C_1_ carbon bonds. Bond was monitored over 5000 AIMD time steps of the AIMD simulation.

Molecular dynamic (MD) simulations were performed using NAMD (NAMD ver. 2.1) [25]. Potential energies were calculated using the CHARMM36 forcefield [26]. Previously published information describes the same methods and conditions used [8]. Briefly, the protonation state for each titratable residue at pH 6.0 was determined using PROPKA [27,28]. Counterions (Na^+^, Cl^−^) were added to a box of TIP3P water molecules to achieve a NaCl concentration of 0.1 M. MD simulations were conducted at 333.15 °K and atmospheric pressure using a Langevin thermostat with periodic boundary conditions [29]. Long-range electrostatic interactions were calculated using the particle Mesh Ewald method and a cutoff distance of 12 Å [30]. The simulation was performed by running a 10 ps energy minimization step, a 10 ps energy minimization step, a 10 ps energy minimization step, and then heating the system to the desired temperature (333.15 °K) over 300 ps. The heated system was then equilibrated at 333.15 °K for 5 ns under an isothermal−isobaric ensemble (NPT). Finally, the system’s 50 ns production simulation was performed in the canonical ensemble (NVT). The flexibility and dynamics of the protein structure and acids were calculated as the per-residue root mean square fluctuation (RMSF) using VMD 1.9.4 software [31].

## 3. Results

### 3.1. Low pH Accelerates Catalysis of Bond Cleavage of Lignin Dimers by Wildtype Laccase from F. mediterranea

Products that resulted from catalysis of four main reactions, β-O-4′ ether bond cleavage, C_α_-C_1_ carbon bond (ring A) cleavage, C_α_-O.H. oxidation, and aromatic ring hydroxylation, by Fom_lac with NIMS tagged GGE and VGE dimers as substrates were quantified using the NIMS assay (Figure 1). The data in Figure 1A,B show that wildtype Fom_lac showed its highest catalytic capability at pH 3.0 for GGE and VGE dimers, with more than 95% of the dimers being converted to products (Figure 1, cyan bars). At pH > 7.0, reaction with wildtype Fom_lac laccase resulted in more than 40% and 95% of the GGE and VGE substrates being unmodified, respectively. At all pH levels, β-O-4′ ether bond cleavage (Figure 1, blue bars) was the dominant catalytic event, followed by C_α_-C_1_ carbon bond cleavage (Figure 1, red bars), C_α_ oxidation (Figure 1, green bars) of the GGE dimer, or aromatic ring hydroxylation of the VGE dimer (Figure 1, purple bars). These pH profiles are consistent with previous studies of laccases from the white rot-causing fungus *Cerrena unicolor* [8] and with reports on the pH effect on the activity of laccases from *Trametes versicolor,* where the authors suggested the pH of the surrounding environment can influence the configuration of the metal ions of the laccase and the redox potential of the copper ions, which, in turn, affects the enzyme’s ability to bind and oxidize substrates [9].

Pham et al. [13] proposed a pH-dependent mechanism for degrading a phenolic lignin dimer degradation and used quantum calculations and AIMD simulations. These calculations were extended to include the Gibbs free energy of reaction (ΔG, kcal/mol) and AIMD simulations of the non-phenolic lignin dimer. The cationic radical intermediate formed from Fom_lac-catalyzed 1-electron oxidation of GGE (Figure 2A) and VGE (Figure 2B) can undergo a variety of reactions such as side-chain oxidation (C_α_ oxidation of GGE dimer or aromatic ring hydroxylation of VGE dimer), C-C bond, and β-O-4′ ether bond cleavage. β-O-’4’ ether bond cleavage required +32.8 kcal/mol for the GGE dimer and +15.6 kcal/mol for the VGE dimer, and the calculated β-O-’4’ bond breaking was more energetically favorable than C_α_-C_1_ carbon bond cleavage in both of the lignin dimer types. The high ΔG (+312 kcal/mol) for C_α_-C_1_ carbon bond breaking also helps explain its products’ lack of detection in the non-phenolic dimer reaction. 

It was reported that that under acidic conditions, the formation of protonated hydroxyl groups on GGE (Figure 3, left panel) drives the reaction to further completion by lowering bond dissociation energies. In these simulations, the β-O-4′ ether bond cleavage frequency for the GGE cationic radical intermediate was 33.66%, but it was only 11.42% for the non-protonated GGE cationic radical (Figure 3A) [13]. The same trend in bond cleavage frequencies was observed from the analysis of AIMD trajectories of the non-phenolic lignin dimer (VGE, Figure 3B). Protonated VGE intermediates at the C_α_-O.H. or C_β_-O.H. positions resulted in a bond-breaking frequency of 26.25% for the β-O-4′ ether bond, which is much higher than the frequency for the non-protonated VGE cationic radical (0.54%).

### 3.2. Enhanced Lignocellulosic Biomass Degradation Was Achieved by Fusing a CBM to the Laccase from F. mediterranea

CBMs are classified based on substrate binding properties and sequence homology reflecting cellulose binding similarities. Molecular dynamic (MD) simulations were performed to screen candidate CBM family 1 domains (fungal cellulases) for the consequent generation of Fom_CBM chimeras. Eleven CBM domains from different fungal cellulases were fused to the C terminal of Fom_lac using the polylinker (ASPPPPTTTTSSAPATTTTAS), and their stabilities were investigated using MD simulations (Figure 4A). These simulations aimed to identify the most stable CBM—Fom_lac complex, so values of per-residue root mean square fluctuation (RMSF) for CBM domains were used to guide the selection of more stable CBM domains. In general, increased flexibility was observed for adjacent residues (aa 450–498) located on the protein surface of the CBM domain, where this region is more likely to be easily disrupted by heat and solvent interactions (Figure 4B). The most stable CBM domain was CBM10 of Exoglucanase 1 from *Trichoderma reesei,* and CBM10 was thus used to construct the Fom_CBM used in further studies and applications in this work. 

Fom_CBM was characterized for its ability to catalyze bond breaking in the phenolic GGE and non-phenolic VGE NIMS-tagged dimers over a range of pH conditions using the NIMS assay. Fom_CBM catalysis of bond cleavage efficiencies in GGE and VGE at pH 2–5 was comparable to that for the wildtype Fom_lac laccase (Figure 5), indicating that the addition of a CBM did not alter the catalytic efficiency of the laccase. The total products from breaking both β-O-4′ ether and C_α_-C_1_ bonds in the presence of the mediator HBT totaled ~80% for phenolic GGE at pH 2.0 and 60% for non-phenolic VGE at pH 2.0. Similar to Fom-lac, the Fom_CMB laccase showed deficient activity towards both of the model dimers at alkaline pH (7–10), even in the presence of HBT.

Adding laccase enzyme mixtures of cellulases and hemicellulases in saccharification reactions represents a potential synergistic approach to more efficiently convert lignocellulosic biomass into fermentable sugars compared to mixtures of just cellulases and hemicellulases. We further hypothesized that adding a CMB to Fom_lac would allow Fom_lac to bind to biomass, position it in closer proximity to lignin, and increase its ability to catalyze the depolymerization of lignin and thus work synergistically with cellulases and hemicellulases to improve glucose and xylose yields. Conversion of dry-milled pine with particle sizes greater than 250 µm resulted in low yields (~3–5%) of fermentable sugars with and without the addition of Fom_lac or Fom_CBM to the Ctec3/Htec3 enzyme mixture (Figure 6A). In the conversion of finely dry-milled pine (particle size < 25 µm), the addition of Fom_lac to the Ctec3/Htec3 enzyme mixture did not show increased sugar yields; when the mediator HBT was added to the mixture, sugar yields increased by 38.9% compared to the Ctec3/Htec3 enzyme mixture alone (Figure 6B). The highest sugar yields were achieved when Fom_CBM was added to the Ctec3/Htec3 mixture, which resulted in a 41.6% increase in sugar yields without HBT and a 140.3% increase in yields when HBT was included in the reaction (Figure 6B). Increased sugar yields were also observed for mixtures of Ctec3/Htec3, Fom_CBM, and HBT in saccharification of [Ch][Lys]-pretreated pine. The highest sugar yield (72% of glucose and 23% of xylose) was observed from the saccharification of [Ch][Lys]-pretreated pine, which showed a 32.5% increase in total sugars compared to the Ctec3/Htec3 mixture without Fom_CBM (Figure 6C). We did not observe a significant yield of low molecular products from lignin degradation after the reaction. Further analysis of the large structure of lignin before and after the reaction is required in future studies. 

## 4. Discussion

The presence of different bond types and the structural heterogeneity and complexity of lignin make it challenging for enzymes to evolve to break specific bonds selectively. Despite these challenges, understanding lignin degradation is crucial for developing sustainable processes for utilizing lignocellulosic biomass in producing bio-derived fuels and chemicals. Advancements in analytical techniques, such as nuclear magnetic resonance (NMR) spectroscopy, mass spectrometry, and high-performance liquid chromatography, have enabled researchers to gain insights into the structure of lignin and the products formed during enzymatic depolymerization of lignin. In this study, the highest yield of products from catalysis of C_α_-O.H. bond, C_α_-C_β_ bond, β-O-4′ ether bond, and C_α_-C_1_ carbon bond cleavage by the laccase from *F. mediterranea* (Fom_lac) was observed at pH 2–3 for both phenolic and non-phenolic lignin dimers as the substrate. Furthermore, data from AIMD simulations and Gibbs free energy calculations performed on protonated hydroxyl lignin dimers showed lower activation energies for bond breaking, which helped explain why the bond-breaking efficiency was much higher at a low pH, especially for the β-O-4′ ether bonds, and results in higher overall conversion yields of dimer degradation. These results again emphasize the important role of low pH conditions in driving reaction equilibrium toward the favorable formation of the active cationic radical intermediate and in controlling bond-cleavage frequencies through the protonation of hydroxyl groups [13]. Because the protonation of lignin drives the reaction, these results provide insights into the optimal design of reaction conditions to improve the efficiency of lignin degradation catalyzed by ligninases and potentially any catalytic approach to lignin depolymerization. 

Fusing laccase with a carbohydrate-binding module (CBM) is a biotechnological strategy to improve the enzyme’s efficiency in lignocellulose and plastic degradation. Laccases are multicopper oxidases capable of oxidizing a variety of phenolic and non-phenolic substrates. At the same time, CBMs are non-catalytic domains that can specifically bind to carbohydrates, facilitating the enzyme’s attachment to the complex structure of lignocellulosic biomass or synthetic polymers [32,33,34,35]. The potential benefit of fussing a CBM onto a laccase was also emphasized in this study, with wildtype Fom_lac catalyzed oxidation of lignin through long-distance electron transfer through an aqueous solution from lignin to mediator (Figure 7A). Adding a CBM to Fom_lac to form Fom_CBM significantly improved fermentable sugar yields from finely dry-milled pine particles and ionic liquid pretreated pine. Our working model is that the CBM provides Fom_CBM with an additional domain that can specifically bind to cellulose, other carbohydrate components, or possibly even lignin in lignocellulosic biomass, resulting in the laccase being closer in proximity to lignin and thus enhancing the required electron transfer between the laccase and lignin. We observed improved laccase activity and sugar yields only in the presence of a mediator. This result further suggested that via the CBM bringing the laccase much closer in proximity to lignin, the electron transfer pathway between the mediator and lignin was significantly shortened, resulting in enhanced lignin depolymerization, improved access to cellulose and hemicellulose, and enhanced enzymatic saccharification (Figure 7B). This approach reflects the ongoing efforts in bioengineering to create tailored enzymes for more efficient and sustainable bioprocessing of biomass. Optimization of the fusion protein and process conditions may be necessary to fully realize the benefits of adding CBMs to laccases and other lignin-degrading enzymes for lignocellulose degradation.

## 5. Conclusions

The results from experimental and computational studies of both the wildtype laccase from *F. mediterranea* (Fom_lac) and an engineered variant containing a carbohydrate-binding module (Fom_CBM) showed that both variants efficiently catalyzed breaking of β-O-4′ ether and C_α_-C1 bonds in phenolic and non-phenolic lignin model dimers. Both Fom_lac and Fom_CBM showed highest activity at an acidic pH (pH = 3 to 4) and in the presence of the reaction mediator hydroxybenzotriole. Further, this work investigated the use of the two laccases to improve the enzymatic saccharification of ionic liquid pretreated pine. The highest sugar yields were detected when Fom_CBM was added to a mixture of cellulases and hemicellulases. Taken together, adding laccases to current cellulase and hemicellulase mixtures can increase substrate accessibilities of hydrolytic enzymes, which is a promising approach to improve lignocellulose degradation and is essential for producing biofuels, biochemicals, and other value-added products.

## Figures and Tables

**Figure 1 biomolecules-14-00324-f001:**
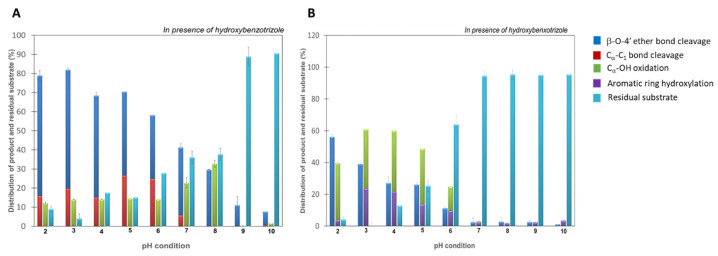
Product distribution from bond cleavage of fluorous-tagged GGE (**A**) and VGE (**B**) by wildtype Fom_lac. The reaction contained 1 mM of NIMS-tagged lignin dimer, 5 µM of Fom_lac enzyme, and 20 mM of 1-hydroxybenzotriole as mediator and was performed in sodium acetate buffer pH 2.0–10.0. Error bars are the standard deviation for three replicates.

**Figure 2 biomolecules-14-00324-f002:**
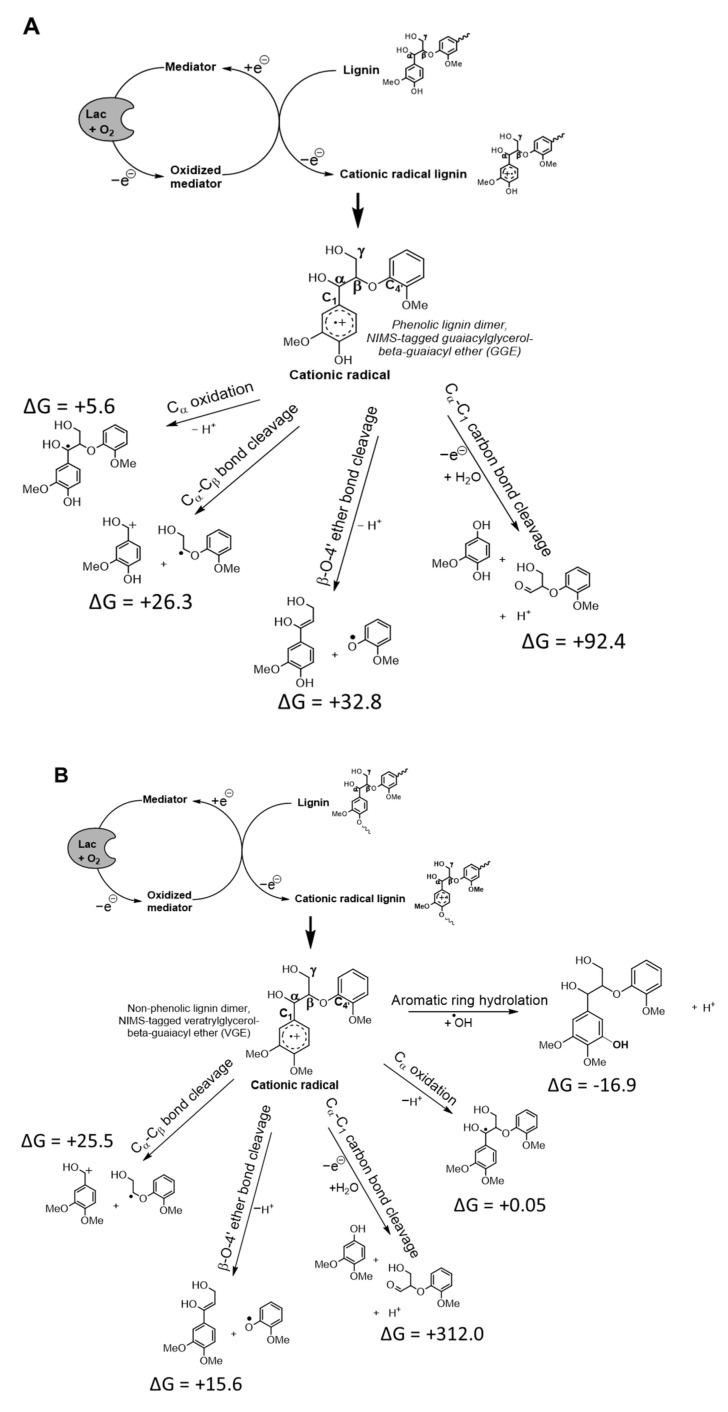
The proposed scheme of Fom_lac laccase catalyzed depolymerization and ΔG. Relative Gibbs free energy of reaction (kcal/mol) was calculated for various bond types via cationic radical intermediate from phenolic lignin dimer (**A**) and from non-phenolic lignin dimer (**B**).

**Figure 3 biomolecules-14-00324-f003:**
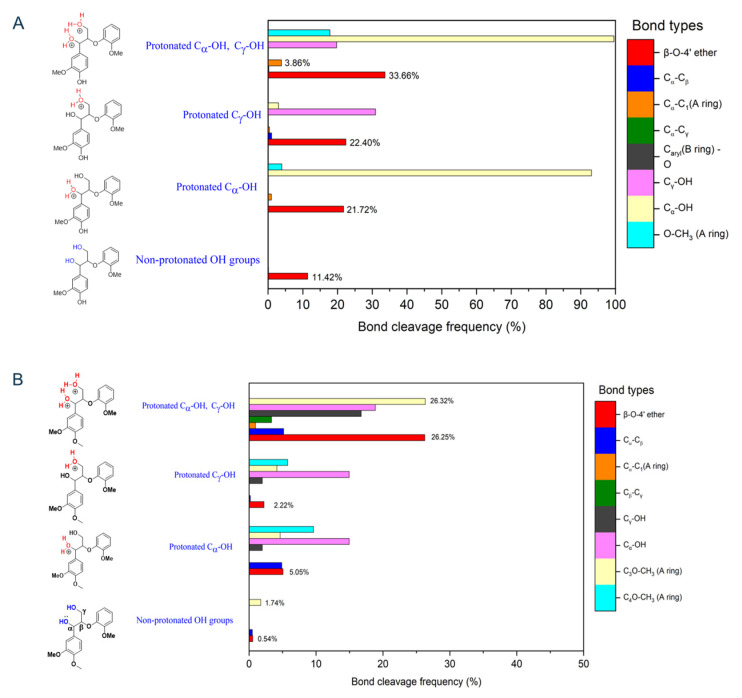
Bond-cleavage frequency computed by AIMD simulation of non-protonated and protonated OHs GGE (**A**) and VGE (**B**) cationic radicals for different linkages.

**Figure 4 biomolecules-14-00324-f004:**
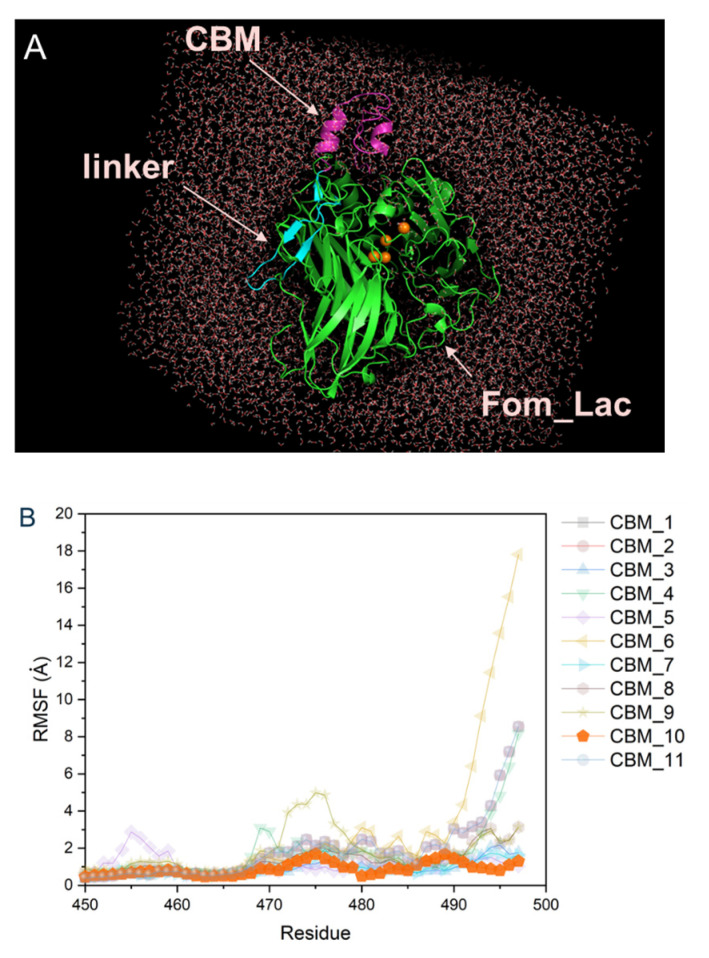
Three-dimensional visualization of Fom_CBM molecule in the solvent box for M.D. simulations (**A**) and per-residue root mean square fluctuation (RMSF) for different CBM domains when fused with Fom_lac through polykinker (**B**). CBM1—Endoglucanase I (gene egl1) from *Trichoderma reesei*; CBM2—Endoglucanase II (gene egl2) from *Trichoderma reesei*; CBM3—Endoglucanase V (gene egl5) from *Trichoderma reesei*; CBM4—Endoglucanase from *Myceliophthora thermophila*; CBM5—Exoglucanase I (gene CBHI) *Trichoderma reesei*; CBM6—Exoglucanase I (gene CBHI) from *Phanerochaete chrysosporium*; CBM7—Exoglucanase I (gene CBHI) *Trichoderma viride*; CMB8—Exoglucanase II (gene CBHII) from *Trichoderma reesei*; CBM9—Exoglucanase 3 (gene cel3) from *Agaricus bisporus*; CBM10—Exoglucanase 1 from *Trichoderma reesei*; CBM11—Exoglucanase 1 from *Humicola grisea var. thermoidea.*

**Figure 5 biomolecules-14-00324-f005:**
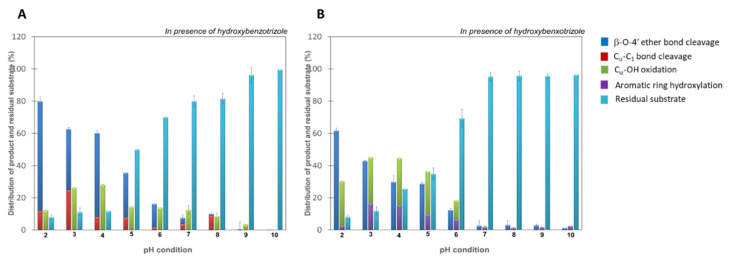
Product distribution from bond cleavage of fluorous-tagged GGE (**A**) and VGE (**B**) by Fom_lac fused with CBM—Fom_CBM. The reaction contained 1 mM of NIMS-tagged lignin dimer, 5 µM of Fom_lac enzyme, and 20 mM of 1-hydrobenzotriole as mediator and was performed in sodium acetate buffer pH 2.0–10.0. Error bars are the standard deviation for three replicates.

**Figure 6 biomolecules-14-00324-f006:**
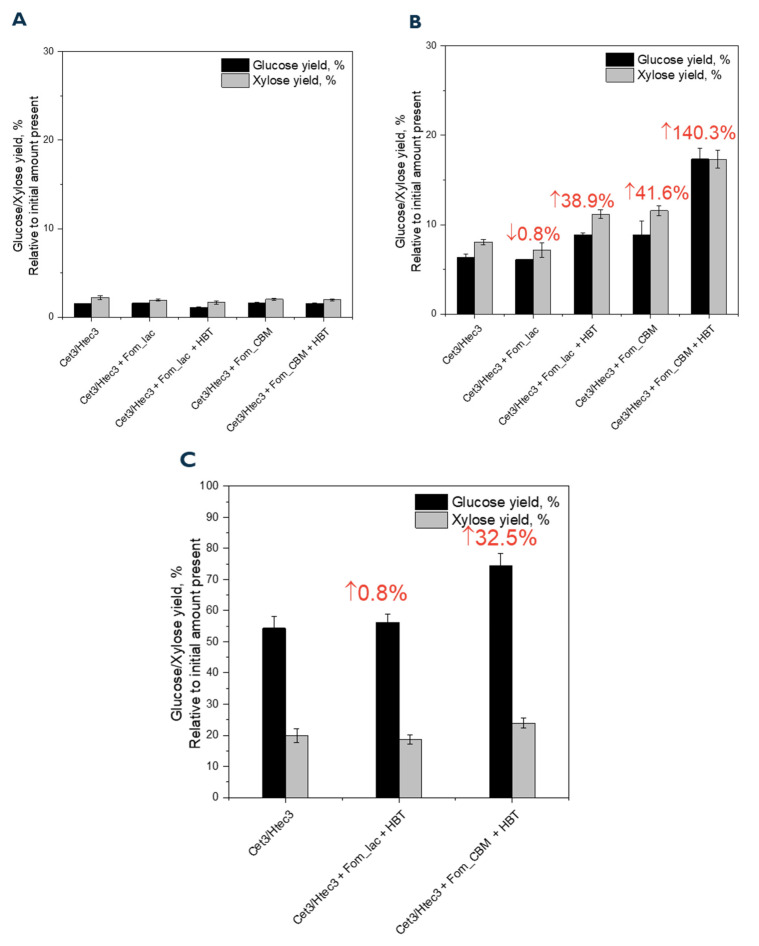
Synergistic effect of laccase and cellulases/hemicellulases in saccharification of different pine biomasses. (**A**) Untreated pine biomass with particle size > 250 µm. (**B**) Untreated pine biomass with particle size < 250 µm. (**C**) [Ch][Lys]-treated pine. Saccharification conditions: loading 2.5 wt%, 10 mg enzyme (CTec3:HTec3, 9:1 *v*/*v*) per g pine, 5 uM For_lac or Fom_CBM, in 0.1 M sodium acetate at pH 5.0, 50 °C, 72 h. Mediator: HBT: 1-Hydroxybenzotriazole (5 mM). Pretreatment conditions: pine (20 wt%), IL (80 wt%), 140 °C, 3 h.

**Figure 7 biomolecules-14-00324-f007:**
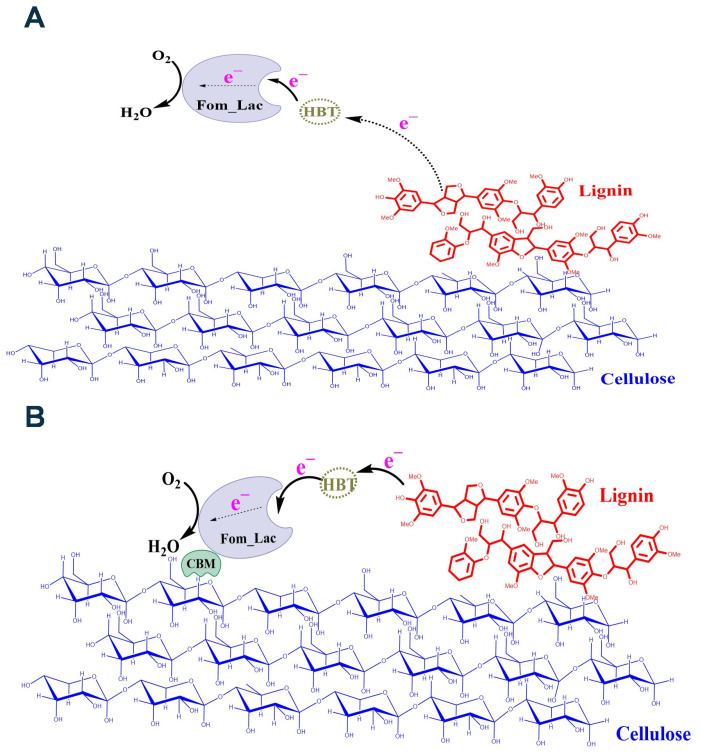
Proposed mechanism of lignin-catalyzed Fom_lac in the presence/absence of CBM. (**A**) Long-distance electron transfer through an aqueous solution is inefficient from lignin to mediator. (**B**) CBM brings Fom_lac in close contact with lignin and shortens the electron transfer pathway between mediator and lignin.

## Data Availability

The datasets used and/or analyzed during the current study are available from corresponding author on reasonable request.

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
