# Peer review of "An Engineered Laccase from Fomitiporia mediterranea Accelerates Lignocellulose Degradation"

_biomolecules, 2024, doi:10.3390/biom14030324_

Round 1

Reviewer 1 Report

Comments and Suggestions for Authors

I have carefully reviewed the manuscript with the title "An engineered laccase from Fomitiporia mediterranea accelerates lignocellulose degradation" submitted to Biomolecules. Overall, the study presents a comprehensive investigation into the laccase from Fomitiporia mediterranea and its catalytic performance, stability, and synergistic effects with cellulases and hemicelluloses. While the study contributes valuable insights, I have identified several areas that require attention before the manuscript can be considered for publication.

1. Choice of Ionic Liquid-Pretreated Pine Biomass:

Clarify the rationale behind choosing ionic liquid-pretreated pine biomass as the experimental material. Explain the advantages or unique features of using this specific biomass in the Introduction or other parts of the study.

2. Lack of Citations for Scientific Facts:

Provide references for scientific facts presented in the manuscript, especially in lines 58-64 and 67-79. Ensure that relevant studies supporting statements are appropriately cited.

3. Structural Issues, Particularly in Introduction:

Address the structural issues in the introduction (e.g., lines 76-79) to ensure a smoother transition between sections. Avoid introducing experimental details prematurely.

4. Clarify "which researcher" in Line 86:

Specify the researcher referred to in Line 86. Provide the name or context to enhance clarity.

5. Clearer Structure, Avoid Redundancy:

Streamline the structure, particularly in lines 107-128, and eliminate redundant information. Ensure that data is presented in a clear and concise manner, avoiding repetition.

6. Provide Details about F. mediterranea Strain:

Include specific information about the F. mediterranea strain, such as strain number or any relevant details. Explain why this particular strain was chosen for the study.

7. Missing References in Materials and Methods:

Ensure that the Materials and Methods section includes appropriate references, particularly in lines 155-166 and 193-204, to support the methodology used.

8. Figure Placement:

Ensure that figures are appropriately placed in the manuscript, and adjust the placement of Figure 1 in line 240 and other relevant sections.

9. Explain How Laccase Degrades β-O-4’ ether bonds and of C–C bonds:

Address the theoretical inconsistency in line 240 where it is mentioned that laccase theoretically cannot degrade β-O-4’ ether bonds and of C–C bonds. Explain how this is possible in the context of the study.

10. Figure 1: Full Names for GGE and VGE:

In Figure 1, provide the full names for GGE and VGE to enhance clarity and understanding.

Please address these comments and make the necessary revisions for the improvement of the manuscript.

Author Response

Please check our responses for your comments/suggestions for our manuscript. 

  1. Choice of Ionic Liquid-Pretreated Pine Biomass:

Clarify the rationale behind choosing ionic liquid-pretreated pine biomass as the experimental material. Explain the advantages or unique features of using this specific biomass in the Introduction or other parts of the study.

Response: Pine is a highly adaptable softwood variety that can grow in many environments. Owing to the high carbohydrate content, pine have been recognized as a potential energy crop (https://www.energy.gov/eere/bioenergy/2016-billion-ton-report). In addition to high carbohydrate content and abundance, pine also has a high lignin content, making it an ideal substrate for our study. Due to its high lignin content, pine is a recalcitrant feedstock for which pretreatment (for example, ionic liquid-based pretreatment) is essential for enhancing enzyme accessibility, and its high lignin content makes it an ideal substrate for studying the utility, in terms of increased sugar yields, of adding lignin modifying enzymes such as laccases to the enzyme mixture.

We chose IL-pretreated pine partly because IL pretreatment is an ongoing area of our research at the Joint BioEnergy Institute (JBEI). However, it is highly effective at solubilizing biomass and producing high sugar yields, but efforts are still underway to improve the pretreatment process, including adding lignin-modifying enzymes to the saccharification step. Thus, while the focus of this study was to carry out fundamental studies to express, purify, and characterize a recombinant laccase from Fomitiporia mediterranea, we also evaluated its utility in terms of enhanced monosaccharide production as a supplemental enzyme during saccharification of pretreated biomass.

We added an abbreviated version of this text to the Introduction at lines 109-112 (highlighted in blue). 

  1. Lack of Citations for Scientific Facts:

Provide references for scientific facts presented in the manuscript, especially in lines 58-64 and 67-79. Ensure that relevant studies supporting statements are appropriately cited.

Response: We added references 3,4 (line 66), ref 4 in line 86

  1. Structural Issues, Particularly in Introduction:

Address the structural issues in the Introduction (e.g., lines 76-79) to ensure a smoother transition between sections. Avoid introducing experimental details prematurely.

Response: We removed "In this study, laccases from the white-rot-causing basidiomycete Fomitiporia mediterranea were heterologously expressed in Komagataella pastoris (formerly Pichia pastoris) and characterized on a variety of phenolic and non-phenolic lignin substrates.". We introduced our study goal from lines 102 - 108

  1. Clarify "which researcher" in Line 86: Specify the researcher referred to in Line 86. Provide the name or context to enhance clarity.

Response: We re-wrote the sentences, line 83 -85

  1. Clearer Structure, Avoid Redundancy: Streamline the structure, particularly in lines 107-128, and eliminate redundant information. Ensure that data is presented in a clear and concise manner, avoiding repetition.

Response: We revised

  1. Provide Details about F. mediterranea Strain: Include specific information about the F. mediterranea strain, such as strain number or any relevant details. Explain why this particular strain was chosen for the study.

Response: Thank you for your comment. However, we could not find the additional information for the F. mediterranea strain; we achieved the laccase protein sequence from the JgI portal (https://mycocosm.jgi.doe.gov/Fomme1/Fomme1.home.html), which was described in the method section, line 128 - 130

  1. Missing References in Materials and Methods: Ensure that the Materials and Methods section includes appropriate references, particularly in lines 155-166 and 193-204, to support the methodology used.

Response: We included the Reference [14] for the purification method, line 150, and Reference [15,16] for enzymatic saccharification reactions, line 175.

  1. Figure Placement: Ensure that figures are appropriately placed in the manuscript, and adjust the placement of Figure 1 in line 240 and other relevant sections.

Response: We checked and adjusted the placement of Figure 1: "Products that resulted from catalysis of four main reactions, b-O-4 ether bond cleavage, Ca-C1 carbon bond (ring A) cleavage, Ca-OH oxidation, and aromatic ring hydroxylation, by Fom_lac with NIMS tagged GGE and VGE dimers as substrates were quantified using the NIMS assay (Figure 1)." – line 236 - 239

  1. Explain How Laccase Degrades β-O-4' ether bonds and of C–C bonds: Address the theoretical inconsistency in line 240 where it is mentioned that laccase theoretically cannot degrade β-O-4' ether bonds and of C–C bonds. Explain how this is possible in the context of the study.

Response: We cited other published papers that reported pH-dependent catalysis by laccases. Line 248-252: " These pH profiles are consistent with previous studies of laccases from the white rot-causing fungus Cerrena unicolor [14] and with reports on the pH effect on the activity of laccases from Trametes versicolor where the authors suggested the pH of the surrounding environment can influence the configuration of the metal ions of the laccase and the redox potential of the copper ions, which, in turn, affects the enzyme's ability to bind and oxidize substrates [37].

We also cited the study of Pham et al. (Ref 18), through Gibbs free energy of reaction (ΔG, kcal/mol) and AIMD simulation, that paper proposed a mechanism relating the formation of protonated hydroxyl groups which assisted bond cleavage at acidic condition (3 – 5).

  1. Figure 1: Full Names for GGE and VGE: In Figure 1, provide the full names for GGE and VGE to enhance clarity and understanding.

Response: We corrected figure 1

Reviewer 2 Report

Comments and Suggestions for Authors

Overall, except for a few typos here and there, the manuscript is written very well. However, the manuscript needs serious attention/revision before it can be published in any peer-reviewed international journal.

Although I have listed most of these concerns in the attached file, I am listing some of them here:

1) There is a disconnect between the theme of the paper and the pH study of the laccases as the authors have failed to connect the pH study with the application of laccase in the saccharification of biomass. It seems that the synthesis, pH study, along simulation part can be published separately.

2) I am not completely convinced that the increase in glucose yields was due to the proximity of CBM-containing laccase to the cellulose unless the authors perform additional characterizations of solid and liquid portions. If this was the case then xylose yields should have gone up too for the CBM_FOM as the removal of lignin would provide more/better/higher access to xylonytic enzymes too. Please see my detailed comment in the attached file.

3) The impact of laccase supplementation on the fermentability of resulting enzymatic hydrolyzate is missing as that would provide a clear picture of whether laccase can be co-utilized with cellulase/hemicellulase.

4) The impact of enzyme and/or laccase loadings would help in determining whether laccase can help in lowering the overall cellulase loading and saving cost.

Comments on the Quality of English Language

Please fix some of the typos noted in the attached file.

Author Response

Please check out response for your comments/suggestions for our manuscript: 

  1. There is a disconnect between the theme of the paper and the pH study of the laccases as the authors have failed to connect the pH study with the application of laccase in the saccharification of biomass. It seems that the synthesis, pH study, along simulation part can be published separately.

Relating comment: I don't seem to understand the relevance of this study for this work? Since the hydrolysis was conducted at pH above 5 then where do you stand on this.

Response: These are excellent points. The emphasis of this study was to characterize the heterologously expressed laccase from F. mediterranea using an assay we previously developed that allows the quantification of bond-breaking events in model lignin compounds under various conditions. The experimental and computational work showed that F. mediterranea laccase has the highest activity in cleavage of both β-O-4' ether bonds C–C bonds under acidic conditions, consistent with our previous studies. We then wanted to go a step further and test the potential of this laccase to facilitate the saccharification of biomass, exceptionally high lignin, and highly recalcitrant biomass (pine, in this case). This was based on a preliminary study we did several years ago but didn't follow up because we didn't have purified laccases (i.e., we only had low purity commercially available laccases with low purity that may have contained cellulolytic enzymes). Given the positive results we saw and wondering if the addition of a CBM to the laccase might allow the complex to bind biomass and potentially bring the laccase in closer proximity to lignin, we then fused a CBM to the laccase to examine the possibility of improving sugar yields even further While these additional studies were not the main focus of the manuscript, we felt they were an exciting finding and valuable addition to the manuscript.

Regarding pH mismatches between Htec/Ctec and the laccase, we chose to run experiments combining these enzymes at the pH optimum for Htec/Ctec. This was based on the Novozymes user manual data showing that their cellulases are inactive at pH < 2 and drop off rapidly for pH < 5.5 (file:///Users/klsale/Downloads/Ctec2.pdf). Our data shows that the laccase maintains high activity at pH 5 and 6, particularly in 1-hydroxybenzotriole (Figure 1)

2. I am not completely convinced that the increase in glucose yields was due to the proximity of CBM-containing laccase to the cellulose unless the authors perform additional characterizations of solid and liquid portions. If this was the case then xylose yields should have gone up too for the CBM_FOM as the removal of lignin would provide more/better/higher access to xylonytic enzymes too. Please see my detailed comment in the attached file.

Related comment: There was no increase in sugar yields for Fom with HBT? How come? Explain? (line 312 – 314)

But did you analyze the solid residue or lignin products in liquid to support. Unless you do this, this is a mere jucnture and doesn't support/justify the claims. Also, I would be concenrned that by adding CBM to laccase wouldn't you be blocking cellulose surface thus reducing cellulase action. Therefore, it is important to include data on the effects of enzymes (cellulase) and laccase loadings on cellulose hydrolysis. Rather, I would perform hydrolysis with pure cellulose to see what kind of impact this supplementation of laccase would have on cellulose hydrolysis.

Response: Thank you for the insightful comments. Our data suggests that HBT and CBM may affect different biomass types (pretreated). First, HBT and CBM did not impact sugar yields for untreated pine biomass with particle size > 250 µm. In this case, the substrate may not be easily accessible to the enzyme's active site due to its insolubility (Figure 6A). Second, HBT and CBM individually improved sugar yields over the native laccasse, but the combination showed no additional improvement on untreated pine biomass with particle size < 250 µm (Figure 6B). Lastly, we only saw improved sugar yields for the CBM-laccase complex compared to the HBT-supplemented samples (i.e., laccase + HBT vs CBM-laccase + HBT) on pretreated pine biomass. This suggests complex interactions among the enzymes, biomass type, pH, and mediator that, while extremely interesting, are challenging to sort out and beyond the scope of this work.

We agree with you that there are a lot of further analyses needed to fully understand the mechanisms by which laccase supplementation to hydrolytic enzymes increases sugar yieldsWe do mention this concern on lines 314 – 317 and are setting up collaborations for these analysis work for both biomass and pure cellulose, which will be scope for further study. There are also interesting questions that could be answered by performing experiments that would allow tracking the proximity of the laccase to lignin and cellulose (e.g., FRET or DEER), but these are well beyond the scope of this work.

3) The impact of laccase supplementation on the fermentability of resulting enzymatic hydrolyzate is missing as that would provide a clear picture of whether laccase can be co-utilized with cellulase/hemicellulase.

Response: Thank you for your comments. This work has encouraged us to co-utilize laccase with cellulase/hemicellulose to improve sugar yields, and, yes, fermentation experiments would add additional information about the production of potential inhibitors. Our goal in this work was to demonstrate improved saccharification yields as part of our enzyme mixture optimization efforts. These enzymes are being utilized in ongoing work focused on so-called "one-pot" processes where pretreatment, saccharification, and fermentation are all performed in a single reactor.  

4) The impact of enzyme and/or laccase loadings would help in determining whether laccase can help in lowering the overall cellulase loading and saving cost.

Response: Thank you for the comments.  – we understand the overall cost concerns that the reviewer may have. Based on our previous experience, we have not seen any significant impact on sugar yields by decreasing the CTec3 amounts. We varied the CTec3/HTec3 ratio from 9/1 through 5/5 and observed no significant differences in yields for an ionic liquid pretreated biomass feedstock (unpublished data). We acknowledge that commercial enzyme cocktails could be expensive; therefore, using other cellulases in combination with laccases is an interesting approach to tackle this. Nevertheless, this would be a research topic on its own and out of the scope of the current study.

Thank you for your comments. This work has encouraged us on the potential for co-utilization of laccases with cellulase/hemicellulose mixtures to reduce enzyme costs. While we didn't do a complete scan of enzyme loading, the saccharification experiments used 10mg of enzyme per g of biomass, only half the enzyme loading for saccharification of ionic liquid pretreated biomass.

Other comments:

  1. Did you run experiments with cellulose + hemicellulase supplemented with HBT mediator alone to see there was any impact of this compund alone.

Response: At the experimental design stage, we were also concerned about the possible inhibitory effects of HBT on the activity of Ctec3/Htec3, and we ran experiments with cellulose + hemicellulase supplemented with HBT mediator alone. We did not observe any inhibitory effect on cellulase/hemicellulase as usually reported for phenols (https://pubmed.ncbi.nlm.nih.gov/34252461/)

  1. IT is VERY VERY crucial that include a set of experiment where an equ. amount of BSA is added in place of laccase and see if this increase in yield is due to lignin blocking or lignin hydrolysis.

Response: Thank you for the comments – This is an excellent suggestion to use BSA as a negative control to exclude the possibility of lignin blocking. We will take note to perform this type of direct control in future work. Our evidence for the laccase not blocking lignin is a bit indirect, but the fact that the addition of laccase to the Ctec3/Htec3 does not increase sugar yields (Figure 6) suggests it is not blocking lignin, else we would have seen increased sugar yields if blocking lignin is an effect that increases sugar yields. In the case of the laccase with fused CBM, the laccase is likely bound to cellulose, yet there is no increase in sugar yields (Figure 6), again suggesting the laccase is not blocking lignin. Nevertheless, we observed increased sugar yields with the two laccases (Fom_laccase and Lac_CBM) in the presence of the HBT mediator, which we have shown directly improves lignin degradation rather than just the occurrence of lignin blocking.

Also, our detailed studies of Fom_laccase and Lac_CBM demonstrated they catalyze the cleavage of β-O-4' ether bonds, Cα-Cβ bonds, and Cα-C1 carbon bonds, all of which are enhanced in the presence of HBT.

Minor suggestions:

  • Provide the protein contents and activity numbers for these two enzyme preps.
  • Didn't use some sort of antifungal or antibiotic agent to arrest the growth of bugs.I would also include details on the sampling time/frequency. Also, it would be helpful to include the formulas you used to calculate yields.
  • Provide the protein contents and activity numbers for these two enzyme preps.

Response: We revised the method section and added additional information as you suggested (blue highlighted, lines 170 - 200)

What was the source of heating for the reactor?

Response: The reactor was heated using a 4871 process controller supplying power to the heating jacket.

How long does it take to heat and cool the reactor? Were these times included in the reaction time?

Response: It takes 45-60 minutes to heat it up and about 30 minutes to cool it down. No, the time was not included in the reaction time.

Round 2

Reviewer 2 Report

Comments and Suggestions for Authors

The authors have only addressed the formatting issues and haven't made any efforts to address the major concerns/issues I raised in my previous report. Thus, I can not recommend it for publication yet.

I still would like to see proper response (with additional experiments/evidence) to the following issues:

1)-

  1. There is a disconnect between the theme of the paper and the pH study of the laccases as the authors have failed to connect the pH study with the application of laccase in the saccharification of biomass. It seems that the synthesis, pH study, along simulation part can be published separately.
  2.  I am not convinced by the authors argument on additional experiments with pure cellulose. I feel it is must that such experiements are done to support the claims that the close proximity of laccase containing CBM to cellulose is the reason for enhanced glucose yields.

Author Response

  1. There is a disconnect between the theme of the paper and the pH study of the laccases as the authors have failed to connect the pH study with the application of laccase in the saccharification of biomass. It seems that the synthesis, pH study, along simulation part can be published separately.

Response: We thank the reviewer for the comment and apologize for the confusion, as we may have inadvertently implied the theme of the manuscript and the work aimed to characterize the laccase's pH profile and then perform experiments with cellulases and hemicellulases at the optimal pH of the laccase. The primary aim of this work was to characterize the catalytic performance of a heterologously expressed (Komagataella pastoris) and purified laccase from the white-rot causing basidiomycete F. mediterranea (Fom_lac) by quantifying β-O-4' ether, Cα-Cβ, and Cα-C1 bond cleavage products, as well as the by-products produced from Cα-oxidation and polymerization of reaction products using a nanostructure initiator mass spectrometry (NIMS) assay with phenolic and non-phenolic lignin-like model compounds [11,12]. Both the effects of pH and the presence of the reaction mediator hydroxybenzotriole (HBT) were investigated. A secondary aim was to evaluate the utility of Fom_lac and an engineered variant of Fom_lac to which a carbohydrate-binding module (CBM) was fused to its C-terminus (Fom_CBM) to enhance saccharification yields from dry milled and cholinium ([Ch]+) lysinate ([Lys]−) ([Ch][Lys]) pretreated pine (Pinus radiata) by measuring glucose and xylose yields. We modified the manuscript to indicate these aims more clearly. We feel that the impact of these combined studies are greater than dividing the manuscript into two smaller papers and that the extension of the fundamental studies of pH tolerance and activity into a real-world demonstration of how this engineered enzyme may improve biomass deconstruction when added to a saccharolytic enzyme mixture would be a valuable addition to the scientific literature.

This was also indicated in our original response:

The emphasis of this study was to characterize the heterologously expressed laccase from F. mediterranea using an assay we previously developed that allows the quantification of bond-breaking events in model lignin compounds under various conditions. The experimental and computational work showed that F. mediterranea laccase has the highest activity in cleavage of both β-O-4' ether bonds and C–C bonds under acidic conditions, consistent with our previous studies. We then wanted to go a step further and test the potential of this laccase to facilitate the saccharification of biomass, exceptionally high lignin, and highly recalcitrant biomass (pine, in this case). This was based on a preliminary study we did several years ago but didn't follow up because we didn't have purified laccases (i.e., we only had low purity commercially available laccases with low purity that may have contained cellulolytic enzymes). Given the positive results we saw and wondering if the addition of a CBM to the laccase might allow the complex to potentially bind biomass and possibly bring the laccase in closer proximity to lignin, we then fused a CBM to the laccase to examine the possibility of improving sugar yields even further. While these additional studies were not the main focus of the manuscript, we felt they were an exciting finding and valuable addition to the manuscript.

Regarding the pH disconnection,  Htec/Ctec and the laccase have different pH optima, and we ran the experiments with mixtures of all enzymes at the pH optimum for Htec/Ctec. This was based on the Novozymes user manual data showing that their Ctec/Htec enzyme activity drops off rapidly for pH < 5.5 and is inactive at pH < 3. Our data shows that the laccase still maintains high activity at pH 5 and 6, particularly in the presence of 1-hydroxybenzotriole (Figure 1), so we performed saccharification at pH 5.5.

  1. I am not convinced by the authors argument on additional experiments with pure cellulose. I feel it is must that such experiements are done to support the claims that the close proximity of laccase containing CBM to cellulose is the reason for enhanced glucose yields.

Response: With our current data, we only suggest that the CBM allows the laccase (Fom_CBM) to bind to the biomass, which brings the laccase in closer proximity to lignin (Lignin is embedded between cellulose, hemicellulose components on biomass structure), facilitating more efficient removal of lignin and better cellulase access to cellulose. We are not suggesting that Fom_lac or Fom_CBM catalyzes the conversion of cellulose into glucose and that bringing the laccase closer to cellulose improves this. We are not convinced that experiments with pure cellulose will help answer the question of whether the CBM allows Fom_CBM to bind cellulose because the experiment's output is glucose yields. Therefore, no change in glucose yields would not conclusively indicate that Fom_CBM doesn't bind to cellulose. Fom_CBM can bind cellulose; however, laccase generally does not catalyze glycosidic bond cleavage. We are working on setting up the method for detection of binding occurrences between carbohydrate-binding modules (CBMs) and cellulose crystals though Surface Plasmon Resonance (SPR), Isothermal Titration Calorimetry (ITC)…We think many other factors still need to be investigated to understand the mechanism of this enzyme cascade. However, those are not the scope of this manuscript. In this manuscript, besides aiming to characterize laccase and its variants on lignin dimeric models degradation, we aim to evaluate the potential application of CBM-fusing laccase and hypothesize that Fom_CBM binds cellulose (and possibly lignin) on insoluble biomass substrate, which promotes more efficient lignin removal and, as a result, increases access of cellulase to cellulose.